# Review of the Current Status on Ruminant Abortigenic Pathogen Surveillance in Africa and Asia

**DOI:** 10.3390/vetsci11090425

**Published:** 2024-09-12

**Authors:** George Peter Semango, Joram Buza

**Affiliations:** School of Life Sciences and Bioengineering, Nelson Mandela African Institution of Science and Technology, Arusha P.O. Box 447, Tanzania

**Keywords:** livestock, ruminants, abortion, Africa, Asia

## Abstract

**Simple Summary:**

Abortions in ruminants lead to economic losses to livestock-owning communities. The major causes of these abortions are infectious agents. Surveillance of the infectious agents that cause abortions is important to the overall improvement of livestock productivity. Most LMICs have scarce or inadequate surveillance platforms for these pathogens. In this review article, we have summarized the current status of the available surveillance platforms in place for the infectious agents that lead to abortions in ruminants, as well as the sero-epidemiology of these pathogens.

**Abstract:**

Ruminant abortion events cause economic losses. Despite the importance of livestock production for food security and the livelihoods of millions of people in the world’s poorest communities, very little is known about the scale, magnitude, or causes of these abortions in Africa and Asia. The aim of this review was to determine the current status of surveillance measures adopted for ruminant abortigenic pathogens in Africa and Asia and to explore feasible surveillance technologies. A systematic literature search was conducted using PRISMA guidelines for studies published between 1 January 1990 and 1 May 2024 that reported epidemiological surveys of abortigenic pathogens Africa and Asia. A meta-analysis was used to estimate the species-specific sero-prevalence of the abortigenic agents and the regions where they were detected. In the systematic literature search, 39 full-text manuscripts were included. The most prevalent abortigenic pathogens with sero-prevalence greater than 10% were BHV-1, *Brucella*, *Chlamydia abortus, Neospora caninum,* RVFV, and *Waddlia chondrophila* in cattle, BVDV in sheep, and RVFV and *Toxoplasma gondii* in goats in Africa. In Asia, Anaplasma, BHV-1, Bluetongue virus, *Brucella,* and BVDV were prevalent in cattle, whereas Mycoplasma was important in goats and sheep.

## 1. Introduction

Abortion is defined as fetal death and expulsion before completion of the pregnancy period in livestock, and can be caused by microorganic abortigenic agents. These disease agents infect the reproductive organs of the animal, resulting in the defective attachment of the fetus and thus its premature expulsion. Abortigenic agents in livestock include bacterial, protozoan, and viral agents [1]. Abortions may also be caused by other factors, such as genetic disorders, trauma, environmental factors such as temperature, nutritional factors such as phytotoxins, including mycotoxins, as well as iatrogenic factors such as the administration of abortigenic drugs [1]. However, infectious agents are the leading cause of abortion in livestock [2]. Common infectious agents that lead to abortion include *Neospora caninum*, *Brucella* spp., and Rift Valley Fever Virus in cattle, *Coxiella burnetii* in goats and sheep, and pestiviruses in sheep [3,4,5,6,7,8]. In South Africa, abortigenic agents reported in resource-poor farmers’ cattle included *Brucella abortus*, *Neospora caninum*, BVD/MD virus, IBR/IPV, *Trichomonas fetus*, and *Campylobacter fetus* [9].

Abortions in livestock are a major cause of economic losses to farmers worldwide, making it an important phenomenon to monitor and control. The magnitude of the economic losses has been quantified in some parts of the world for specific pathogens. For example, in South America, the annual losses due to Neosporosis for the dairy industry were estimated to be USD 43.6 million (range, USD 15.62–194.41 million) in Argentina and USD 51.3 million (range, USD 35.8–111.3 million) in Brazil [10,11]. In Tanzania, gross direct economic losses were recently estimated to be USD 263 million [12].

In addition to economic losses, some livestock abortigenic agents are zoonotic making them relevant to human health as well. Some of those infectious agents, including *Brucella abortus*, Rift valley fever, *Toxoplasma gondii*, and *Campylobacter*, among others, can cause fever and abortion in humans.

Developed countries have been successful in the control of some abortigenic agents by devising and implementing surveillance systems. These surveillance systems capture abortion events as quickly and accurately as possible [13]. The implemented surveillance systems include the use of statutory testing, as well as mandatory reporting by farmers of any abortion event to a veterinary inspector by phone, who would then respond and act accordingly by testing and implementing appropriate interventions [13]. (http://www.gov.scot/Topics/farmingrural/Agriculture/animal-welfare/Diseases/disease/Brucellosis/Surveillance, accessed on 22 May 2022). The surveillance platforms that have been successfully implemented in developed countries include passive, active, targeted, sentinel, syndromic, reportable disease, abattoirs and slaughter slab, and emerging disease surveillance platforms [13]. These surveillance systems are implemented on a regular basis and, for their successful implementation, are coupled with well-trained and equipped response personnel on the ground and state-of-the-art testing facilities [14].

However, it has been noted that one of the major constraints for the control of abortigenic agents in low- and middle-income countries (LMICs) is the absence of qualitative and quantitative information. This is mainly due to the lack of adequate implementation of surveillance systems for livestock diseases in most African and Asian countries. Currently available disease information is dependent on active disease searching by researchers and includes limited or passive participation by the community. This has led to poor control of disease pathogens, including abortigenic agents, in LMICs, leading to unknown economic losses, as well as a lack of guidance for appropriate interventions. In East Africa as a region, there are ongoing research efforts to unravel the epidemiology of disease pathogens, including abortigenic pathogens. As in other LMICs, East African countries have a high burden of abortigenic agents [3,15,16,17,18], but few studies have attempted to estimate the economic losses due to abortions. Currently, the surveillance data collected are not being sufficiently used in rapid response and priority setting in Tanzania [19]. This is mainly because the national surveillance system is not functioning optimally, as in many other LMICs [19,20,21]. This has thus led to massive underreporting of abortion events in Tanzania, whereby approximately less than 10% of all cases are reported (personal communication). The actual causes of the surveillance system not functioning optimally are also undocumented. Typically, the abortion surveillance system requires the abortion events to be reported to the government by the livestock keepers to the Livestock Field Officers (LFOs), stationed at the village level. From there, it is then reported to the District Veterinary Officer (DVO), who reports to the Zonal Veterinary Centre Director (ZVC). The ZVC then informs the Director of Veterinary Services (DVS) at the Ministry Level. Regular reports of the number of abortion events are then provided to the global platforms at the World Organization for Animal Health (OIE). The system is paper-based from the LFO up to the ZVCs.

Despite the availability of established and successfully implemented surveillance systems in place in certain Northern countries, these may not be directly replicable in many African and Asian countries. Indeed, these systems may not be practical due to their financial, infrastructural, and expert requirements. In India, there is an animal disease surveillance program that is limited to a few diseases, such as *Brucella* and *Leptospira*.

Using a systematic literature review process, we assessed the available literature on studies that reported livestock abortigenic organisms in Africa and Asia. We appreciate the large heterogeneity between the regions within and between Africa and Asia, but would like to document the distribution of abortigenic pathogens, especially in the wake of the ongoing climate change and its impact on pathogen distribution. Additionally, we determined the surveillance systems that are being used in Africa and Asia in the reporting of livestock abortion events. There is also a wide disparity between the countries within and between Africa and Asia, especially in terms of socio-economic status, but this also shows the importance of the pathogens in these regions. The main objective of this study was to identify the circulating abortigenic pathogens in Africa and Asia through the sero-prevalence surveys conducted in these respective countries and also document on the surveillance platforms in place for their monitoring.

## 2. Methods

### 2.1. Study Design and Systematic Review Protocol

References were sought and identified following the Preferred Reporting Items for Systematic Reviews and Meta-Analyses (PRISMA) guidelines [22] (Appendix A). Studies were searched in PubMed, Scopus, Embase, and Google Scholar published between 1 January 1990 and 1 May 2024. The search terms are listed in Table 1.

### 2.2. Search Strategy

Article titles and abstracts were screened for suitability for inclusion by GS. Full-text articles were included once the abstracts passed the initial screening. They were selected for full-text review if the studies investigated any of the abortigenic pathogens of interest, reported on samples collected from cattle, goats, or sheep, involved surveillance of the abortigenic pathogens, and data collection took place in African or Asian regions or countries as defined by the United Nations (UN) statistics division [23]. Full-text articles were reviewed independently by two authors (GS and JB) to determine if each article met the pre-determined inclusion and exclusion criteria (Appendix A). Articles were included for full-text review if the full-text article could be retrieved, if it reported primary data, if the article reported surveillance data in sheep, goats, and sheep, regardless of the laboratory methods used, and if the prevalence of abortigenic pathogens could be calculated from the information available in the paper using any sample type.

### 2.3. Exclusion Criteria

(i)If the numerator (i.e., number positive) and denominator (i.e., number tested) information were not reported at the species and sample type levels;(ii)If they were in a language other than English. When required, a third reviewer (TK) served as a tiebreaker, independently reviewing articles to resolve disagreements between the two primary reviewers.

### 2.4. Article Selection and Data Extraction

From each included article, we extracted information on the species of the affected animal, sample type, the total number of samples tested, and total positive samples. The number of pathogens detected was extracted to determine pathogen prevalence. Sample location data included UN statistics division African and Asian geographic region countries [23]. A formal bias assessment was performed (Appendix A), assigning low (L), moderate (M), and high (H) to each potential introduction of bias. The bias elements considered in the formal assessment were related to abortigenic pathogens of interest, studies conducted out of Africa and Asia, and technologies used. An overall assessment of low, moderate, or high risk of bias was assigned to each included article.

### 2.5. Analysis

Prevalence estimates were calculated from pooled data for each pathogen by livestock species and geographic region. Briefly, all the positive cases were summed up as the numerator and all the tested animals were summed up as the denominator, and a pooled sero-prevalence was calculated as a percentage. Summary statistics were calculated in the R program.

## 3. Results

The literature search from the two scientific databases resulted in 297 studies, which included abstracts, free full text, full text, books and documents, clinical trials, and randomized controlled clinical trials including citations. After removing duplicate articles from the searches, 277 articles were available for title and abstract screening. Of these, 57 (20.6%) were identified as potentially relevant and 39 (14.1%) were eligible for inclusion after full-text review (Figure 1). The majority of the studies (18 (46.2%)) were on *Brucella* spp., whereas 9 were on Rift Valley Fever Virus (23.1%), 7 were on *Coxiella burnetii* (17.9%), and 6 each were on *Neospora caninum* and BVDV (15.4%), as summarized in Table 2. The number of studies from each country and the animal species investigated are listed in Table 3.

Two studies (5.1%) of the thirty-nine included were embedded in the national surveillance programs of the respective countries in which they were conducted, South Korea and South Africa, whereas the majority (94.9%) were stand-alone cross-sectional studies. Most studies (30 (76.9%)) were reported from Africa and 9 (23.1%) were conducted in Asia.

Table 2 shows a summary of the information extracted from the 39 full-text articles included from the literature search in the two databases. We extracted information on the country where the study was conducted, year of publication, species from which samples were collected, the number of positive samples among the total number of samples tested, the pathogen detected, the type of animal husbandry method of the species tested, the type of study, as well as the diagnostic method used.

### Median Sero-Prevalence of Abortigenic Pathogens

The adjusted median prevalence calculations estimated Brucella in Africa at 21.5% in 372,127 cattle, and 0.27% and 0.87% in sheep and goats, respectively. *Coxiella* was estimated at 13.0%% in cattle, and 2.3% and 4.5% in sheep and goats, respectively. The most prevalent abortigenic pathogens with sero-prevalence greater than 10% were BHV-1, *Brucella, Chlamydia abortus, Neospora caninum,* RVFV, and *Waddlia chondrophila* in cattle, BVDV in sheep, and RVFV and *Toxoplasma gondii* in goats in Africa. In Asia, Anaplasma, BHV-1, Bluetongue virus, *Brucella*, and BVDV were prevalent in cattle, whereas *Mycoplasma* was important in goats and sheep. Other pathogens detected with low sero-prevalence were *Anaplasma*, BVDV, *Campylobacter, Listeria*, and *Salmonella* in cattle, *Brucella* and *Coxiella burnetii* in goats and sheep, BVDV and *Neospora caninum* in goats, and *Chlamydia pecorum*, RVFV, and *Toxoplasma gondii* in sheep in Africa. In Asia, pestiviruses (BVDV) were prevalent in goats, and *Coxiella burnetii* and *Neospora caninum* were prevalent in cattle. These data are summarized in Table 3.

## 4. Discussion

In this systematic literature search, we found that livestock abortigenic pathogens are still are burden in the livestock sector in African and Asian countries. The most important abortigenic pathogens identified included *Brucella* spp., BHV-1, *Chlamydia* spp., *Neospora caninum*, and *Waddlia chondrophila* in African cattle. RVFV was found to be important in both African cattle and goats. *Toxoplasma gondii* and pestiviruses (BVDV) were important in African goats and sheep, respectively. As for Asia, *Anaplasma*, BHV-1, Bluetongue virus, *Brucella* spp., and BVDV were important in cattle, and *Mycoplasma* was important in sheep. *Brucella*, BHV-1, and pestiviruses were important in both Asia and Africa, while *Anaplasma*, Bluetongue, and *Mycoplasma* were important in Asia only. *Brucella* research in goats and sheep in Asia seems to be minimally conducted, probably because of the ongoing surveillance activities aimed at the pathogen, for instance, in India [20]. Similarly, for *Leptospira,* we could not find a study that detected the pathogen in both Africa and Asia, but it is also under constant surveillance in India [20]. The presence of this surveillance program in India and other parts of Asia may also explain the lack of studies on pathogens such as *Brucella* spp. in goats and sheep. *Brucella* spp. have been reported to be prevalent in India, and our finding of 14.4% sero-prevalence is similar to a recent meta-analysis that reported a pooled sero-prevalence of 16.6% in cattle (O’Donovan & Bersin, 2015) [60]. Anaplasmosis is prevalent in Asia, with different countries reporting different rates of occurrence, such as Iran (37.3%) [61], which is similar to our pooled sero-prevalence. As for Bluetongue virus, which is endemic to Asia and Africa, there is a lack of published data on it in Africa, as also stated elsewhere [62], with very few African countries reporting its occurrence. However, in Asia, BTV has been documented to occur as it is endemic to the region. There are also consortia conducting research on the pathogen, such as the research efforts in Indonesia and Malaysia in collaboration with Australia [63].

It is worth noting that certain pathogens have not been reported at all in Asia in sheep and goats while being present in Africa, such as *Brucella* spp., BVDV, Chlamydia, Coxiella, Neospora, and RVFV. This is unexpected, as the Asia Pacific region hosts over 49% of goats and 22% of sheep in the world [64]. For RVFV, this virus has not yet been spread to most parts of Asia, which explains the lack of published data from Asia [65]. On the other hand, *Mycoplasma, Trypanosoma*, and *Trichomonas* are not as well documented as other pathogens. This may be due to their low sero-prevalence or due to them being neglected as there are other more prevalent pathogens.

Furthermore, surveillance systems for livestock abortigenic pathogens in many African and Asian are so far not optimal, with the exception of a few countries, such as South Africa, India, South Korea, India, and Saudi Arabia which have been reported to have National surveillance programs. The African and Asian regions have the highest rate of growth in surveillance systems using mHealth technology in human medicine. Additionally, most studies employed the serological surveillance approach at single timepoints using a cross-sectional study design. These studies demonstrated the burden of abortigenic pathogens, but were not embedded in the national surveillance systems which would provide continuous real-time information, except for a few Asian countries; namely India, South Korea, Saudi Arabia, and South Africa, which have national Brucella surveillance programs.

In the included articles, most studies used serological tests for pathogen detection. These are cheaper and form a good basis for pathogen monitoring programs compared with molecular diagnostic methods and pathogen culture and isolation. For near-real-time surveillance, serological methods are very useful tools [66].

The establishment of effective surveillance systems for zoonotic diseases has been on the research agenda for some time. This is because it is estimated that 75% of human epidemics and 60% of human pathogens are of animal origin. These facts demonstrate the importance of the surveillance of zoonotic pathogens, among which abortigenic agents belong. These abortigenic agents also cause economic losses in instances where they may not have caused disease to a human.

Several different modes of surveillance have been proposed for zoonotic pathogens in different settings of the world, with varying successes. For instance, in France, it is mandatory for livestock keepers to report abortion events to the veterinary department by calling, and failure incurs a fine of EUR 1500 [67]. However, even with advanced response systems in place in France, there are still many keepers who do not report abortions [67].

Participatory systems using mobile phones have been implemented for veterinary surveillance systems in several countries and across a range of diseases. For example, in Cambodia and Madagascar, participatory surveillance systems using mobile phone technologies have been successfully implemented for the surveillance of animal diseases in remote environments [68].

In Tanzania, as in most other African countries, mobile-based technologies have been trialed in both human and veterinary medicine. Mobile phone technology has been applied successfully in zoonotic diseases, like rabies, in some parts of Tanzania [69]. Other veterinary programs whereby mhealth has been used include the innovative Smartphone App (AfyaData) for Innovative One Health Disease Surveillance from Community to National Levels in Africa [70]. This program has highlighted that rural areas have the potential to utilize mobile phones to link livestock keepers with veterinary professionals and provide timely access to information to assist in the diagnosis and treatment of livestock diseases. Furthermore, the availability of mobile phones in rural areas, in combination with supporting infrastructure and facilities in urban areas, has the potential to stimulate local development and improve the delivery of animal health and extension services [71].

In human medicine, mhealth has been applied more extensively and has been more acceptable among health workers than in veterinary disease surveillance [72]. A number of programs are currently ongoing at the national level. These mhealth programs include maternal health and nutrition programs [73] for HIV/ AIDS [72], Malaria [74], and other diseases. Tanzania is reportedly setting the stage at the global level in integrating eHealth as a component of the national health system. Tanzania has established a community of practice working group since 2009 and in 2011 also developed a National mhealth strategy.

The documented major drawbacks of mobile-based technologies include unclear benefits, uncertain long-term results [73,75,76,77,78,79], and unknown cost-effectiveness [73,80]. Furthermore, there are still issues of under-reporting [77,78]. However, even with the drawbacks, mhealth is by far the most promising surveillance method, especially for zoonotic diseases, and especially in Tanzania with the increasing mobile network coverage and mobile phone ownership in both rural and urban areas. Most developing countries where feasibility studies for the application of mhealth and ehealth have been conducted have reported that most mhealth programs are implemented in silos without the involvement of key stakeholders and hence unsustainability of the mhealth programs [81,82]

We believe that our review has some major strengths in terms of outlining the abortigenic pathogens in ruminants that are found in Africa and Asia; however, our manuscript has limitations in that we were limited to a few databases and also may have not explored all factors and variables that may influence pathogen distribution. However, these were not the immediate objectives of this review.

## 5. Conclusions

In conclusion, livestock abortigenic pathogens are prevalent in many African and Asian countries. Adequate near-real-time surveillance systems for livestock abortigenic pathogens in many African and Asian countries are not present, except for a few countries, such as India, South Africa, South Korea, and Saudi Arabia, which have surveillance programs.

## Figures and Tables

**Figure 1 vetsci-11-00425-f001:**
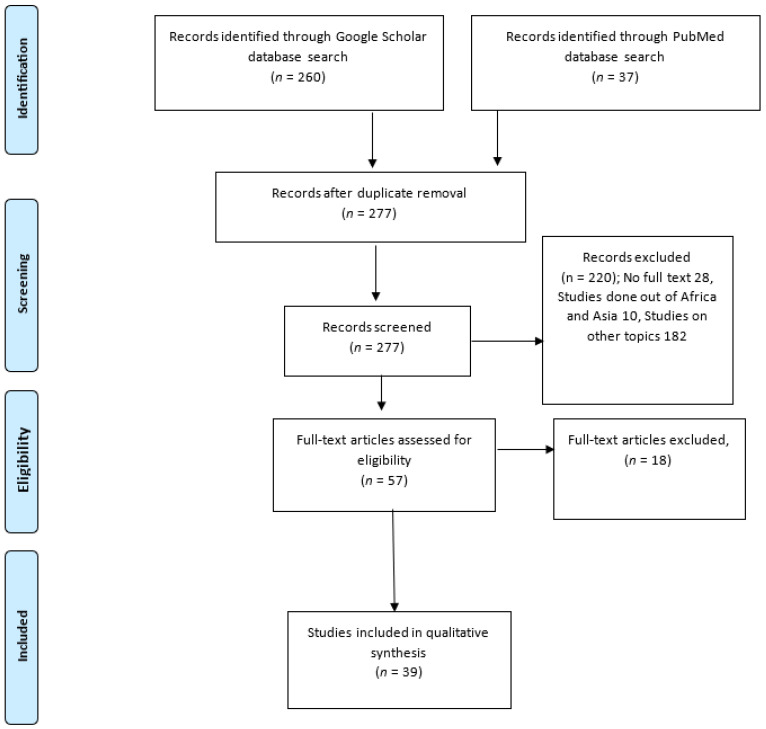
PRISMA flow diagram showing identification, screening, and selection of eligible articles for inclusion in the systematic review, 1990–2002.

**Table 1 vetsci-11-00425-t001:** Literature search strategies.

Search String	Database or Further Sources	Results	Date	Comments
((((ASIA[Text Word]) OR (AFRICA[Text Word]) AND (1990/1/1:2024/5/1[pdat])) AND (((GOATS[Title/Abstract]) OR (SHEEP[Title/Abstract])) OR (CATTLE[Title/Abstract]) AND (1990/1/1:2024/5/1[pdat]))) AND (ABORT*[Title/Abstract] AND (1990/1/1:2024/5/1[pdat]))) AND (surve*[Title/Abstract])	PubMed	37	1 May 2024	PubMed search
abortion surveillance cattle OR sheep OR goats * * * * “Asia OR Africa” -human -people -persons -man -woman -Europe -americas -australia -pacific -“south america”1990–2024	Google Scholar	240	1 May 2024	Google Scholar search through NM-AIST

**Table 2 vetsci-11-00425-t002:** Summary of the information extracted from the full-text articles that were included in the meta-analysis.

s/no	Country	Region	Year	Species	Number of Species (Positive)	Pathogen(s) Detected	Study Type	Husbandry Method (Climatic Zone)	Detection Method	Ref
1	Tanzania	East Africa	1996	Cross-bred bulls; Taurine breeds [24] (Friesian, Ayrshire, and Simmental crossed with Tanzanian short-horn zebu, boran, and Sahiwal)	*Campylobacter fetus* 3/58, *Trichomonas foetus* 0/58	*Campylobacter fetus* subsp. *Venerealis*, *Trichomonas foetus*	Sero-survey	Smallholder dairy farms (zero-grazing) Tropical climate	Culture and biochemical tests	[24]
2	Uganda	East Africa	2000	Cattle (Ankole, crosses—Fresian and Boran)	*Brucella*—41/143 *Anaplasma* 3/454	*Brucella*, *Anaplasma*	Cross-sectional	Pastoral communities Tropical climate	RBPT, ELISA	[25]
3	India	Asia	2002–2004	Cattle	35/427 (9.6%)	*Neospora caninum*	Cross-sectional survey	Dairy farms Tropical climate	ELISA, IFAT	[26]
4	Senegal	West Africa	2003	Sheep	7/260 (2.7%)	RVFV	Serological survey	Nomadic Tropical climate	Sero neutralization test	[27]
5	Zimbabwe	Southern Africa	2004–2005	Cattle	71/1291 (5.5%)	*Brucella*	Cross-sectional	Smallholder Subtropical climate	RBT, ELISA	[28]
6	Sudan	Central Africa	2005	Sheep and goats	Sheep 3/270 (1.1%)	RVFV	Sero-surveillance	Nomadic pastoralist Tropical savannah	ELISA, Hemagglutination	[29]
7	South Africa	Southern Africa	2006–2016	193 cattle, 39 goats, and 57 sheep	63/288; Brucella 21/288 (7.3%) Cattle, *Trueperella pyogenes 5*/*288* Cattle, *1*/*288* sheep	*Brucella, Trueperella pyogenes*, *E. coli, Salmonella*, *L. monocytogenes*, *C. burnetii*, *B. licheniformus*, *Rhizopus*, *B. abortus, Leptospira*, *C. pecorum*, *Campylobacter*	Observational retrospective study	Archived samples Subtropical and temperate	Microbiology, necropsy, histopathology, PCR	[30]
8	Ethiopia	East Africa	2008–2009	Sheep and goats	0/270 sheep, 2/230 goats	*Brucella*	Cross-sectional	Mixed farming Tropical	Rose Bengal Plate Test, Complement Fixation Test	[31]
9	Ethiopia	East Africa	2009–2010	Cattle	2/370 (0.05%)	Brucella	Cross-sectional survey	Mixed farming Equatorial rainforest, Afro-alpine	Rose Bengal, Complement Fixation Test	[32]
10	Tunisia	North Africa	2010–2012	Cattle	214 blood, vaginal swabs, milk. *Brucella* 47/150 (31.3%) RBPT, DANA PCR 46/150 (30.6%). *Chlamydia* 27/150 (18%), *L. monocytogenes* 7/150 (4.6%), *Salmonella* 5/150 (3.3%). Vaginal swabs; *Brucella* 46/150 (30.6%), *Chlamydiales* 27/150 (2.65%), *L. monocytogenes* 4/150 (2.6%)	*Brucella*, *Chamydiales (C. abortus*, *C. pecorum)*, *Listeria*, *Salmonella*, *Coxiella burnetii*, *Campylobacter*	Cross-sectional survey	Limited pasture or tethered Mediterranean climate	PCR, Rose Bengal	[33]
11	Mozambique	Southern Africa	2010–2016	Cattle, goats, and sheep	Cattle 149/404 Goats 45/223 Sheep	RVFV	Sero-survey	Mixed farming Tropical to sub-tropical	ELISA, PRNT	[34]
12	Zimbabwe	Southern Africa	2011	Cattle (mixed breeds)	81/1440 (5.6%)	*Brucella*	Cross-sectional survey	Smallholder, mixed farming (strictly separate pastures) Subtropical	ELISA	[35]
13	Iran	Asia	2011–2012	Sheep and goats	PCR: Sheep 101/274. Goats 10/25, Culture Sheep 76/274. Goats’ 9/25	*Mycoplasma* spp.	Cross-sectional	Mixed farming Arid and semi-arid climate	PCR, bacterial culture	[36]
14	India	Asia	2012–2014	Cattle	11/61 (18.03%)	*Trypanosoma evansi*	Sero-survey	Mixed farming. Tropical climate	ELISA	[37]
15	South Korea	Asia	2012–2013	Cattle (Holstein breed)	37/171 and 85/466	Blue Tongue Virus	Serological survey from National Surveillance Program	Mixed farming Temperate climate	ELISA, BTV neutralization test, RT-PCR	[38]
16	Cameroon	West Africa	2013	Cattle	117/1498	RVFV	Cross-sectional survey	Pastoralists Humid and Equatorial climate	ELISA	[39]
17	Tanzania	East Africa	2013–2016	Cattle, goats, and sheep	*Brucella* Cattle 1/71, *Coxiella* Goats 5/100, Sheep 1/44, *Neospora* Cattle 9/71, Goats 1/100, *Toxoplasma* Sheep 1/44, BHV-1 Cattle 2/49, BVDV Cattle 2/71, Goats 1/100, Sheep 6/44, RVFV Cattle 14/71	*Brucella*, *Chlamydia abortus*, *Coxiella burnetii*, *Leptospira hardjo*, *Neospora caninum*, *Toxoplasma gondii*, Bluetongue Virus, Bovine Herpes Virus 1, Pestiviruses (BVDV/BDV), RVFV	Cross-sectional survey	Pastoral, agro-pastoral, and smallholder Tropical climate	ELISA, PCR	[40]
18	South Africa	Southern Africa	2013–2018	Cattle	359,026 (22.1%)	*Brucella*	Cross-sectional survey, Provincial surveillance program	Mixed farming Subtropical and temperate	CFT, Rose Bengal Plate Test	[41]
19	Mozambique	Southern Africa	2014	Goats	Serology: 31/127 (24.4%)	RVFV	Outbreak investigation	Mixed farming Tropical to sub-tropical	ELISA, PCR	[42]
20	India	Asia	2014	Cattle	160 RBPT 3/160 (1.8%), Standard Tube Agglutination Test (STAT) 5/160 (3.13%)	*Brucella*	Sero-epidemiological survey	Mixed farming Tropical climate	RBPT, STAT, Bacterial culture, Milk Ring Test	[43]
21	Nigeria	West Africa	2015	Cattle	11/97 (11.3%)	RVFV	Cross-sectional survey	Nomadic pastoral Tropical monsoon climate, tropical savannah, and Sahelian hot and semi-arid	ELISA	[18]
22	Kenya	East Africa	2016	Cattle	100/955, 10.5%	*Coxiella burnetii*	Cross-sectional survey	Mixed crop-livestock Tropical climate	ELISA	[4]
23	Egypt	North Africa	2016–2018	Cattle	165/176 (93.86%)	BHV-1	Transboundary, Import from Sudan	Nomadic Subtropical desert climate	ELISA	[44]
24	Tajikistan	Central Asia	2016	Cattle	570 (58 PCR, 12 ELISA)	*Brucella*	Sero-prevalence	Smallholder Continental, subtropical, desert	ELISA, qPCR, DNA sequencing	[45]
25	Tunisia	North Africa	2017	Cattle and sheep	Cattle *Waddlia* 12/27, Parachlamydiaceae 8/27, Chlamydiaceae 7/27, Sheep P. acanthamoebae 9/164, C. pecorum 6/164	*Waddlia chondrophila*, *C. abortus*, *C. pecorum*	Cross-sectional survey	Smallholder Mediterranean	PCR	[46]
26	Algeria	North Africa	2017–2019	Atlas brown cows	650 pregnant (235(36.2%))	*Neospora caninum*	Sero-prevalence	Smallholder Mediterranean	ELISA	[47]
27	Tanzania	East Africa	2017–2019	Cattle	14/63 (23%)	RVFV	Prospective cohort	Pastoral, agropastoral, and smallholder Tropical climate	ELISA, RT-qPCR	[48]
28	Benin	West Africa	2017	Sheep and goats	Goats 83/153, Sheep 3/215	*Toxoplasma gondii*	Sero-epidemiological survey	Pastoral. Steppe climate and topical humid climate	ELISA	[15]
29	Guinea	West Africa	2017–2019	Cattle, goats, and sheep	*Brucella*; Cattle 52/463, Sheep 2/486. *Coxiella*; Cattle 95/463, Goats 18/408, Sheep 11/486. RVF; Cattle 76/463, Goats 4/408, Sheep 5/486	*Brucella*, *Coxiella burnetii*, RVFV	Sero-survey from archived samples	Intensive farms Samples from different prefectures Hot and humid	ELISA, Virus Neutralizing Ab	[49]
30	Algeria	North Africa	2018–2019	Cattle	201/460 (43.7%)	Bovine Herpes Virus 1	Abortion investigation	Mixed farming Mediterranean climate	ELISA	[50]
31	Saudi Arabia	Asia	2018–2020	Sheep and goats	Goat 3/84 (3.5%) Serum	BVDV	Sero-prevalence-Abattoir surveillance	Abattoir, semi-closed management Desert climate	ELISA	[51]
32	Ethiopia	East Africa	2018–2019	Cattle cross and pure breeds; Boran–Fresian cross, Boran–Jersey, Pure Jersey, and Boran	BHV-1 68/86(79.1%), BVD 33/86 (38.4%), *Neospora* 3/86 (3.5%), *Coxiella* 1/86 (1.2%)	*Brucella* spp., *Neospora caninum*, BVD, BHV-1, *Coxiella burnetii*	Reproductive problem investigation	Semi-intensive farming system (grazing and indoor feeding) Equatorial rainforest, Afro-alpine	ELISA	[52]
33	Ethiopia	East Africa	2018–2019	Cattle (Zebu, Holstein, Fresian, and crossbreed)	0/882 (ear notch samples	BVDV	Cross-sectional survey	Peri-urban dairy farms, mixed crop–livestock farms (small holder extensive management system), pastoral herds (seasonal mobility) Equatorial rainforest, Afro-alpine	ELISA	[53]
34	India	Asia	2019	Cattle crossbreeds, exotic, and indigenous	BHV-1 425/1004, BVDV 604/1004, *Brucella* 176/1004, *Coxiella* 57/1004, *Anaplasma* 363/1004, *Neospora* 9/1004	BHV-1, BVDV, *Brucella, Coxiella burnetii*, *Neospora caninum*, *Anaplasma marginale*	Cros-sectional	Intensive dairy farms Tropical	ELISA	[54]
35	Nigeria	West Africa	2020	Cattle	61/1810 (3.37%)	*Brucella*	Cross-sectional	Mixed farming Tropical monsoon climate, tropical savannah, and Sahelian hot and semi-arid	SAT	[55]
36	Egypt	North Africa	2020	Cattle	*Neospora* 35/116 (30.17%), BVDV 31/116(26.72%)	*Neospora caninum*, BVDV	Cross-sectional	Medium-sized farms Subtropical desert climate	ELISA	[56]
37	Kenya	East Africa	2020–2021	Cattle	6593(449)	*Brucella*	Sero-prevalence	Agro-alpine, high and medium potential, semi-arid, arid, and very arid Tropical climate	ELISA	[57]
38	Bangladesh	Asia	2023	Cattle (local, cross)	66/386 (17.09%)	*Brucella*	Cross-sectional	Transboundary area, mixed farming Humid, warm climate	RBPT, Plate agglutination test, serum agglutination	[58]
39	South Africa	Southern Africa	2023	Cattle	2% 770	*Brucella*	Cross-sectional survey, abattoir survey	Communal, commercial, and non-commercial farms Subtropical and temperate	RNT, CFT, Milk Ring Test	[59]

**Table 3 vetsci-11-00425-t003:** Seroprevalence of abortigenic pathogens by species.

Abortigenic Pathogen	Species	Cases (n)		Total Tested (N)		Median Sero-Prevalence	
		Africa	Asia	Africa	Asia	Africa	Asia
*Anaplasma*	Cattle	3	363	454	1004	0.7	36.2
BHV-1	Cattle	436	245	771	1004	56.5	24.4
Bluetongue virus	Cattle	0	122	0	637	0	19.2
*Brucella* spp.	Goats	2	0	230	0	0.87	0
	Sheep	2	0	754	0	0.27	0
	Cattle	80,165	305	372,127	2120	21.5	14.4
BVDV	Goats	1	3	100	84	1	3.6
	Sheep	6	0	44	0	13.6	0
	Cattle	66	604	1155	1004	5.7	60.2
*Campylobacter*	Cattle	3	0	58	0	5.2	0
*Chlamydia abortus*	Cattle	34	0	177	0	19.2	0
*Chlamydia pecorum*	Sheep	6	0	164	0	3.7	0
*Coxiella burnetii*	Goats	23	0	508	0	4.5	0
	Sheep	12	0	530	0	2.3	0
	Cattle	196	54	1504	1004	13	5.4
*Listeria*	Cattle	7	0	150	0	4.7	0
*Mycoplasma*	Goats	0	10	0	25	0	40
	Sheep	0	101	0	274	0	36.9
*Neospora caninum*	Goats	1	0	100	0	1	0
	Cattle	282	44	923	1431	30.6	3.1
RVFV	Goats	80	0	758	0	10.6	0
	Sheep	15	0	1016	0	1.5	0
	Cattle	381	0	2596	0	14.7	0
*Salmonella*	Cattle	5	0	150	0	3.3	0
*Toxoplasma gondii*	Goats	83	0	153	0	54.2	0
	Sheep	4	0	259	0	1.5	0
*Trichomonas foetus*	Cattle	0	0	58	0	0	0
*Trypanosoma evansi*	Cattle	0	11	0	61	0	18
*Waddlia chondrophila*	Cattle	12	0	27	0	44.4	0

## Data Availability

Dataset available on request from the authors.

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
