# Peer review of "Review of the Current Status on Ruminant Abortigenic Pathogen Surveillance in Africa and Asia"

_vetsci, 2024, doi:10.3390/vetsci11090425_

Round 1

Reviewer 1 Report

Comments and Suggestions for Authors

Comments on the Quality of English Language

The quality of writing is fair

Reviewer 2 Report

Comments and Suggestions for Authors

Authors presented a manuscript regarding a revision of some abortogenic diseases surveillance in Africa and Asia. 

In my opinion, the manuscript has serious flaws to be accepted in this journal: 

1) The experimental design (systematic review) missed to include the last years (just up to 2021), missed to include IBR, for example, among other important pathogens, such as Trichomonas or Campylobacter. Moreover, the region they decided to compare is extremely huge (Africa and Asia) with very different handlings, breeds, climates, etc. 

2) They included a quite limited number of studies than the obtained by the research. Moreover, table 2 presented important mistakes, such as total amount of Toxoplasma or BVDV. Additionally, table 3 showed a huge amount of samples that, I guess, they merged from various studies... It should be done by cohorts and detailedly explained.

3) They do not comply with the format policiy of this journal: abstract is too long and semeed to be just copied from the previous journal, they missed keywords and simple summary, bibliography is wrongly cited throughtout the text and in the bibliography section, among others. 

Therefore, I recommend not to accept this manuscript in this journal.

Comments on the Quality of English Language

minor

Round 2

Reviewer 1 Report

Comments and Suggestions for Authors

The manuscript has greatly improved and now provides sufficient detail. There are some minor typographical errors that require the authors' attention. Additionally, pathogen names that need to be italicised should be, and the missing details for sections such as author contributions, informed consent statement, conflict of interest, etc should be provided.

Comments on the Quality of English Language

The manucript can now be accepted.

Reviewer 2 Report

Comments and Suggestions for Authors

As i said in my previous revision: authors presented a manuscript regarding a revision of some abortogenic diseases surveillance in Africa and Asia.  They included some of my suggestions and I am commenting them throughout my previous revision and their answers.

In my opinion, the manuscript has serious flaws to be accepted in this journal: 

1) The experimental design (systematic review) missed to include the last years (just up to 2021), missed to include IBR, for example, among other important pathogens, such as Trichomonas or Campylobacter. Moreover, the region they decided to compare is extremely huge (Africa and Asia) with very different handlings, breeds, climates, etc. 

Response#1: We have re-done our literature search to include upto 2024. We have included all the pathogens that were reported. We have done some re-arrangements to have studies done from Asia separately presented from African studies in Table 2. We also have now included IBR as BHV-1 as well as Trichomonas and Campylobacter.

The main issue in this topic is the experimental design they proposed. I think it does not make sense to compare and describe this among all of these areas. They have different handling conditions, management practices, weather, breeds, etc. All of this variables does interfere on the results. Although they included studies up to 2024, the objective of the work is not correct.

Moreover, comparing this version with the previous one, they included more pathogens but reduced the sources search (no scopus nor emabse results). Additionally, they reduced the N of studies included. 

2) They included a quite limited number of studies than the obtained by the research. Moreover, table 2 presented important mistakes, such as total amount of Toxoplasma or BVDV. Additionally, table 3 showed a huge amount of samples that, I guess, they merged from various studies... It should be done by cohorts and detailedly explained.

Response#2:We have re-done the literature search in 2 databases (PubMed and Google Scholar), we no longer have access to Embase and SCOPUS. We have re-done tables 2 and 4. Table 4 provides the pooled samples sero-prevalence in African and Aisian cohorts separately.

I just commented above about this issue.

3) They do not comply with the format policiy of this journal: abstract is too long and semeed to be just copied from the previous journal, they missed keywords and simple summary, bibliography is wrongly cited throughtout the text and in the bibliography section, among others. 

Response#3: Thank you for the observation. We have made changes to the manuscript including shortening the abstract, including keywords and adjusting the the bibliography to the journal specifications.

 Authors still without accomplishing the MDPI guidilenis, nor Veterinary Sciences recommendation: abstract still being super long, bibliography is not correct throughout the text nor in the references section, tables are not appropriate, they did not complete the final requests of the manuscript (author contribution, funding, etc.).

Therefore, I recommend not to accept this manuscript in this journal.

In conclusion, I still thinking that the experimental design is not appropriate at all and the manuscript is not acceptable to be published and shared with the Academia. The main idea of authors could be interesting, but how they designed this review is not. I encourage them to keep improving by continents, include more variables and compare data appropriately. Otherwise, these results are not valid.

Comments on the Quality of English Language

minor
